# Manipulation and control of droplets on surfaces in a homogeneous electric field

Johannes Hartmann [1,2], Maximilian T. Schür [1,2] & Steffen Hardt [1✉]

A method to manipulate and control droplets on a surface is presented. The method is based on inducing electric dipoles inside the droplets using a homogeneous external electric field. It is shown that the repulsive dipole force efficiently suppresses the coalescence of droplets moving on a liquid-infused surface (LIS). Using a combination of experiments, numerical computations and semi-analytical models, the dependence of the repulsion force on the droplet volumes, the distance between the droplets and the electric field strength is revealed. The method allows to suppress coalescence in complex multi-droplet flows and is real-time adaptive. When the electric field strength exceeds a critical value, tip streaming from the droplets sets in. Based on that, it becomes possible to withdraw minute samples from an array of droplets in a parallel process.

[1] Technische Universität Darmstadt, Fachbereich Maschinenbau, Fachgebiet Nano- und Mikrofluidik, Alarich-Weiss-Straße 10, D-64287 Darmstadt, Germany.
[2] These authors contributed equally: Johannes Hartmann, Maximilian T. Schür. ✉email: hardt@nmf.tu-darmstadt.de

In the past two decades, droplet microfluidics has experienced a rapid development. There are two competing versions of droplet microfluidics. First, the creation and processing of droplets may occur inside microchannels. After many of the corresponding fundamentals had been studied[1], the number of applications relying on channel-based droplet microfluidics[2] has been increasing continuously over the past few years. Second, there are open droplet microfluidic platforms. In such devices, droplets are transported and manipulated on planar substrates. In that context, droplet manipulation based on electrowetting-on-dielectric[3,4], surface acoustic waves[5] and dielectrophoresis[6] has been reported.

The control of droplet coalescence is a major challenge of droplet microfluidics. On the one hand, it can either be desirable to merge specific droplets, for example to mix liquids and let them react. On the other hand, when handling a large number of droplets in parallel, it is often imperative to suppress coalescence, simply to keep the compartmentalization of liquid into small distinct spaces intact. The control of droplet coalescence becomes comparatively easy when a method allowing to locally manipulate individual droplets is available, which, however involves complex system architectures[4]. In most other systems, however, the key is to suppress undesired droplet coalescence, which is usually done by adding surfactants to one of the two immiscible phases[7]. These surfactants can have a number of undesired effects. When droplets are used as reaction spaces for the synthesis of specific materials, surfactants introduce contaminations that compromise the purity of the reaction products. For these reasons, efforts were made to establish surfactant-free versions of these processes. Examples are the surfactant-free synthesis of nanoparticles based on droplet flow inside a capillary[8] and the surfactant-free synthesis of nanoparticles[9,10] or Janus particles[11] using a microfluidic droplet generator. Furthermore, surfactants can have detrimental effects on biomolecules contained in the droplets. Generally, biomolecules in droplets can lose their activity when interacting with surfactants[12]. In that context, it has been shown that surfactants influence enzyme activity[13] and protein expression[14]. Therefore, for a number of applications it would be preferable to eliminate the need of surfactants.

While in droplet microfluidics it is desirable to have a base configuration ensuring that two approaching droplets do not merge, at the same time it is desirable to induce the coalescence of specific droplets in a controlled manner. Inducing droplet coalescence in microfluidic devices has been in the focus of intense research efforts. Droplets can be merged based on electrostatic stresses acting on the liquid–liquid interface[15,16]. Coalescence can be induced by tailoring the microchannel geometry, for example by introducing a sudden opening of the flow path[17,18], or by introducing pillar-shaped structures[19]. It was shown that thermocapillary convection can support coalescence[20,21]. Coalescence can be prompted by bringing two droplets into contact at a channel junction[22,23]. Bringing subsequent droplets in an extensional flow can serve to merge the droplets[24]. Tailor-made droplet manipulation based on surface acoustic waves is another method to induce coalescence[25]. It has also been studied how coalescence can be promoted by tuning the surfactant concentration at the liquid–liquid interface[26,27].

Another challenge related to droplet microfluidics is the addition and extraction of minute volumes to or from droplets. In context with channel-based droplet microfluidics, the sample-transfer problem has been solved by bringing a droplet in touch with a liquid meniscus at a microchannel junction[28], a process that can be supported by an electric field[29,30]. An alternative solution is based on the avoided coalescence of a droplet with a liquid reservoir under an electric field[31]. For open droplet microfluidics, the problem of subtracting minute samples from

droplets has only been solved in a rudimentary manner. This problem arises, among others, when the goal is to couple droplet microfluidics to analytical protocols based on mass spectrometry. In that context it was demonstrated how samples may be withdrawn from droplets on electrowetting-on-dielectric arrays[32]. The solution in context with electrospray ionization mass spectrometry is to transport the droplets to dedicated sites where field-induced tip streaming is initiated. In a broader context, however, comparatively few reports exist on minute samples being withdrawn from sessile droplets. One recent example is the pinch-off of small daughter droplets from a parent droplet based on electrowetting driven by high-voltage AC signals[33].

In this work we consider an open microfluidic device in which droplets move along a liquid-infused surface (LIS)[34], which means that aqueous droplets are separated from a solid surface by a thin oil film. The droplets are exposed to a homogeneous electric field normal to the surface. Based on that, we demonstrate schemes for droplet manipulation and control. We show that the electric field serves to suppress droplet coalescence, whereas without electric field, rapid coalescence is observed. Further, we demonstrate that with sufficiently strong electric fields, withdrawing minute samples from arrays of droplet is possible in a parallel process. These operations are based on a generic electrostatic scheme, with the voltage as external control parameter, and are largely independent of surface chemistry or native charges, in contrast to other principles of electrostatic droplet manipulation[35,36]. The motivation for considering droplets on a LIS is twofold. First, owing to the low contact-angle hysteresis of droplets on LIS, their motion along the surface gives reliable and reproducible information about the forces acting on them. Second, suppressing the coalescence of droplets on LIS is especially challenging, since the oil wetting ridge surrounding the droplets[37–39] induces short-range attractive forces that promote coalescence[40–42]. Apart from that, in the past few years controlled droplet manipulation on LIS was demonstrated[40,43–45], making such systems open droplet microfluidic platforms in their own right.

## Results

**Experimental design**. The main focus of the experiments is to characterize the repulsion between two deionized (DI) water droplets induced by a homogeneous electric field. The droplets pass each other on an slightly inclined LIS. A schematic of the conducted experiments can be found in Fig. 1. To let the droplets pass each other in an controlled manner, one droplet is immobilized on the LIS by a single pinning site. The moving droplet is applied through a nozzle. Subsequently, the droplet slides down the inclined surface, driven by gravity. As the driving force is counteracted by the droplet's drag force on the LIS, the droplet reaches a steady sliding velocity after a short time. The nozzle is arranged in such a way that coalescence can be expected when the moving droplet approaches the immobilized droplet. The coalescence event can be prevented by applying a homogeneous electric field normal to the LIS which is realized by applying a voltage between the LIS and an indium tin oxide (ITO)-coated glass cover.

**Droplet trajectories and friction law**. Figure 2a shows a superposition of four images from a typical experiment from which the droplet trajectory can be inferred. As the sliding droplet approaches the immobilized one it is pushed away from the latter. However, this only occurs if a sufficiently high voltage is applied between the electrodes as indicated by the Supplementary Movies 1 and 2. Consequently, we hypothesize that the acceleration of the droplet originates from an electrostatic repulsion force $\mathbf{F}_r$ between

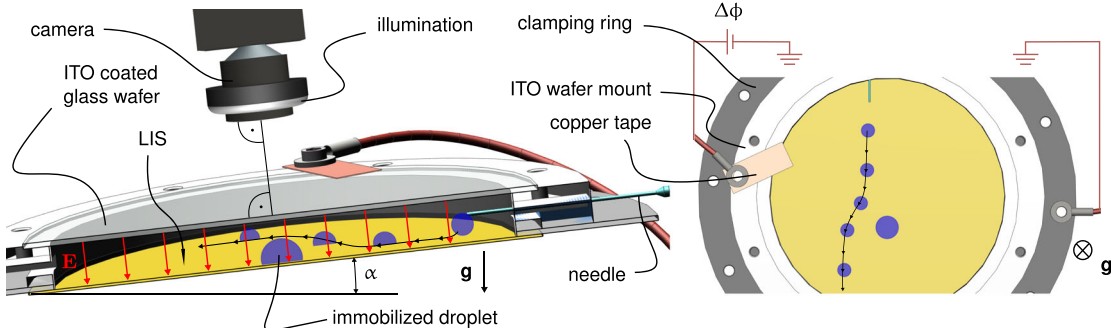

**Fig. 1 Experimental setup.** On the left-hand side of the figure, a schematic of the air-filled parallel-plate capacitor containing the droplets is shown. An electric field, visualized by red arrows, is applied between an indium tin oxide (ITO) layer on a transparent fused silica wafer, facing towards the camera, and the bottom side of a lubricant-impregnated silicon wafer. While the immobilized droplet is deposited on the surface prior to the experiments, the mobile droplets are pinched off from a nozzle during the measurements. As indicated by the sketched trajectory (black line), the gravity-driven mobile droplet is repelled from the immobilized droplet due to electrostatic interactions. The trajectories of the droplets are recorded with a camera. On the right-hand side of the figure the camera view of a droplet interaction is shown schematically.

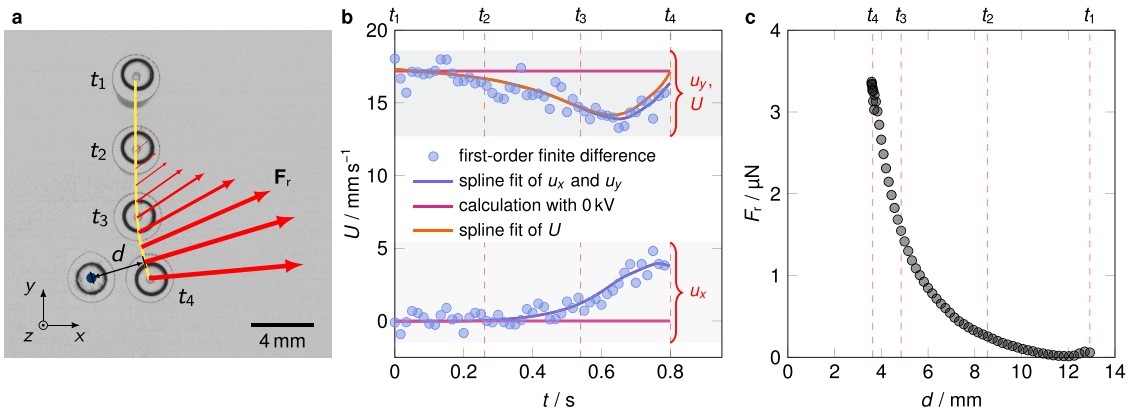

**Fig. 2 Example of a single experiment and its analysis.** The adjustable volumes $V_1$ and $V_2$ of the two droplets are 10 and 10.25 µL, respectively. A potential difference of 4.5 kV is applied between the electrodes. In all plots $t_1$, ..., $t_4$ refer to the same instances in time. **a** Superposed images showing an individual mobile droplet (right) passing by an immobilized droplet (left) at four different instances $t_1$, ..., $t_4$. The smoothed trajectory extracted from the recorded image series is plotted as a solid yellow line, whereas the position of the static, immobilized droplet is marked with a blue circle. At certain points of the trajectory the repulsion force $\mathbf{F}_r$ calculated with Eq. (1) is visualized with an arrow. Their length and thickness scale linearly with the force magnitude. The maximum force magnitude displayed here is about 3.4 µN. Owing to measurement uncertainties the repulsion force is not perfectly aligned with the distance vector $\mathbf{d} = d\,\hat{\mathbf{d}}$ connecting the midpoints of the droplets' projection. The scale bar provides a reference for the size of the droplets. **b** Velocity of the moving droplet as a function of time obtained from the recorded frames. Velocities calculated using first-order finite differences applied to the raw data are symbolized by filled markers. Velocities obtained by applying time derivatives to the spline-fitted trajectory are plotted using solid, blue and orange lines. Magenta solid lines represent the computed velocities assuming that no electrostatic repulsion forces would act on the moving droplet. It becomes apparent that the moving droplet is accelerated in x-direction and decelerated in y-direction when it approaches the immobilized droplet. **c** Magnitude of the electrostatic repulsion force $F_r$ between two droplets as a function of their mutual distance, as obtained from the smoothed velocity curves in **b** using Eq. (1).

the droplets. Let $\mathbf{a} = (\ddot{x}, \ddot{y})^T$ and $\mathbf{u} = (u_x, u_y)^T = (\dot{x}, \dot{y})^T$ denote the planar acceleration and velocity of the droplet with mass $m = \rho\,V$, where $\mathbf{x} = (x, y)^T$ is the position of the center of mass, $\rho$ the the mass density and $V$ the liquid volume. Then Newton's second law governing the motion of the sliding droplet can be written as

$$m\,\mathbf{a} = m\,\mathbf{g} - \eta\,U^{-\frac{1}{3}}\mathbf{u} + \mathbf{F}_r\,, \qquad (1)$$

where $U \equiv |\mathbf{u}| = \sqrt{u_x^2 + u_y^2}$. The projected gravitational acceleration is given by $\mathbf{g} = (0, g\sin\alpha)^T$ with the angle of inclination $\alpha$. The second term on the right-hand side immediately follows from the scaling law for the drag force of a droplet on a LIS. In several publications the drag force on a droplet was reported to scale as $F_\eta \propto U^{\frac{2}{3}}$ within a certain range of sliding velocities[38,46–49].

In our studies we use a LIS produced according to the method reported in ref. [50]. We have verified the mentioned friction law for the conditions of our experiments (for details see the Supplementary Methods). The drag coefficient $\eta$ needs to be known to determine the repulsion force from the droplet trajectories and Eq. (1). We calculate $\eta$ individually for each single experiment by measuring the steady sliding velocity $U$ the mobile droplet reaches before it interacts with its immobilized counterpart. In this case the force balance of Eq. (1) simplifies to a balance of gravitational and drag forces, reading $\eta = U^{-\frac{2}{3}}\,m\,g\,\sin\alpha$. Subsequently, we use the smoothed velocity values $u_x$, $u_y$ and $U$ and acceleration values $A \equiv |\mathbf{a}|$, obtained via time derivatives of the spline-fitted trajectories, as suggested in[51], for solving Eq. (1). Exemplary velocity curves are shown in Fig. 2b. This yields $\mathbf{F}_r$ and its magnitude $F_r \equiv |\mathbf{F}_r| = \sqrt{F_{r,x}^2 + F_{r,y}^2}$ which is exemplarily shown in Fig. 2c as a function of the distance between two droplets. We observe that

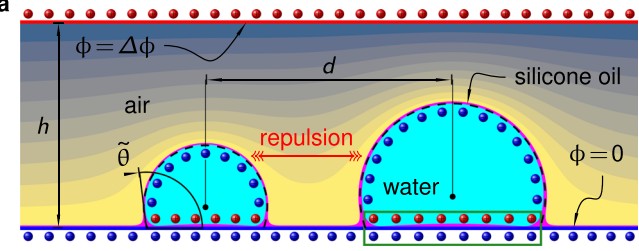

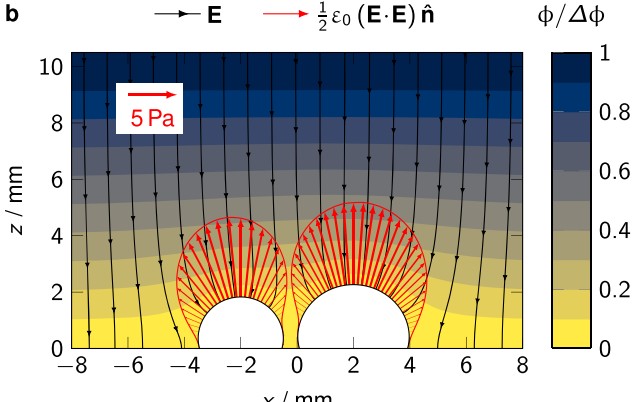

**Fig. 3 Origin and magnitude of electrostatic repulsion.** Schematic illustration of **a** presents electrostatic problem and **b** exemplary numerical solution. **a** Vertical cross section through the droplets (cyan) along the distance vector $\mathbf{d} = d\ \hat{\mathbf{d}}$. A thin lubricant film (magenta) cloaks the droplets and forms a small wetting ridge. Between the lower electrode (blue) and upper electrode (red) a potential difference $\Delta\phi > 0$ is applied. The effective interfaces of the idealized droplets considered in the simulations are symbolized by dashed black lines. The corresponding apparent contact angle is denoted as $\tilde{\theta}$. Red and blue shaded spheres symbolize positive and negative induced surface charges, respectively. The induced surface charges on either side of the idealized footprint, which are encircled by a green rectangle, just compensate each other from a macroscopic point of view. **b** Cross section (xz-plane cutting through the centers of the droplets) showing the numerically computed normalized electrostatic potential $\phi/\Delta\phi$ (colormap) and electric field lines. The applied voltage difference $\Delta\phi$ is 4.5 kV, the droplets have volumes of $V_1 = 9.5\ \mu L$ (left) and $V_2 = 20\ \mu L$ (right). The lateral distance between their centers of mass $d$ is 6 mm. The red arrows represent the Maxwell stress acting on the liquid surface. Their length and width scales linearly with the magnitude of the stress. In the upper left corner, a scale for the arrows is provided. The plot shows only a small fraction of the actual computational domain that extends further in lateral direction.

the inertial term can be omitted from the balance in Eq. (1). The neglect of capillary interaction forces in Eq. (1) can be justified by comparing the effective range of these forces and the minimal measured distance between the droplets, as discussed in the Supplementary Methods.

**Origin of the repulsion force.** Since long-range interaction of the droplets is only observed if a sufficient voltage is applied, we can attribute the repulsion unambiguously to the Coulomb interaction of surface charges induced by the external electric field, as sketched in Fig. 3a. For droplets with finite conductivity (electrolyte solutions) the effective surface charge can be decomposed into contributions from free and bound surface charges. On the molecular level an electric double layer with finite extent will establish at the surface of electrolyte droplets. Since the thickness of this layer, characterized by the Debye length, is several orders of magnitude smaller than the dimensions of the droplets, it is

sufficient to lump the microscopic charge distribution in an effective surface charge as long as we are only interested in the macroscopic repulsion force $F_r$. Expanding the charge distribution sketched in Fig. 3 in multipoles results in a dipole as the leading-order contribution. In the framework of electrostatic field theory the microscopic attractive or repulsive Coulomb interactions between individual surface charges translate into a mechanical stress, the electrostatic Maxwell stress $\frac{1}{2}\varepsilon_0(\mathbf{E} \cdot \mathbf{E})\hat{\mathbf{n}}$, acting normal on the droplet interface $\partial\Omega_d$ with outward-pointing normal vector $\hat{\mathbf{n}}$. Consequently, integrating the Maxwell stress over the droplet surface and projecting the resulting force $\mathbf{F}_r$ onto the normalized distance vector $\hat{\mathbf{d}} = \mathbf{d}/d$ yields the net repulsion force

$$F_r = \hat{\mathbf{d}} \cdot \oint_{\partial\Omega_d} \frac{1}{2}\varepsilon_0(\mathbf{E} \cdot \mathbf{E})\hat{\mathbf{n}}\ dA. \tag{2}$$

Since the fluid surrounding the droplets is uncharged, the electric field is obtained from $\mathbf{E} = -\nabla\phi$ and $\nabla^2\phi = 0$. We note that the expression of the Maxwell stress in Eq. (2) implies that the droplet is conducting or its relative permittivity approaches infinity, which is a reasonable assumption in the present case. Following the arguments presented in previous work[43], the thin lubricant film that cloaks the droplet and forms a wetting ridge around the footprint is ignored in our electrostatic model since its contribution is marginal. Instead the cloaking film is adsorbed in an effective interface and the wetting ridge is collapsed into a three-phase contact line (TCL), characterized by an apparent contact angle $\tilde{\theta}$[37,43].

We have performed static three-dimensional numerical computations of the repulsion force between two droplets. For the purpose of illustration, Fig. 3b shows the electrostatic potential, electric field lines and Maxwell stress computed for an exemplary choice of the model parameters. At a closer look one can notice that the distribution of Maxwell stress is slightly asymmetric with respect to the axis of rotation of each droplet (see also Supplementary Fig. 5). At the side that faces the other droplet the stress is smaller than at the other side. It is this asymmetric distribution that eventually gives rise to a net repulsion force. Assuming perfectly conducting droplets, as done here, yields a upper bound for the repulsion force. By contrast, a lower bound for the repulsion force is found by assuming non-conducting droplets with a specific electric permittivity. The repulsion force between two real droplets will range between these two ideal limits. In the Supplementary Methods both limiting cases are compared numerically. In summary, for aqueous droplets the difference is negligibly small when compared to the experimental uncertainties. Thus we conclude that in case of aqueous droplets the repulsion mechanism is virtually independent of the actual ionic strength.

It is worth noting that since no special measures were taken to prevent spontaneous charging, we must assume that the droplets in our experiments might carry a significant net charge before being deposited in the plate capacitor. As a matter of fact, droplets of DI water can easily acquire net charges of around 0.1 nC by conventional pipetting or similar dispensing methods[52]. Corresponding Coulomb forces between two droplets in free space would be of the same order of magnitude as the repulsion force measured in our experiments. However, as already mentioned, the droplets neither attract nor repel each other if no external electric field is applied. This apparent contradiction can be resolved by considering that the lubricant film is vanishing small compared with the finite electrode spacing. Consequently, any excess charges within the droplet are located right at the footprint of the droplet, where they are compensated by their mirror charges in the lower electrode from a macroscopic point of

view. The neutralization of excess charges at the footprint is analogous to that of the surface charges induced by the external electric field, which are encircled by a green rectangle in Fig. 3a. Eventually, the measured repulsion force is independent of any native net charges and we cannot draw any conclusion about the latter from experimental observations. Analogously, any charge carriers released from potential surface reactions would be shielded in the same way as initial net charges and do not alter the mechanism presented here.

While the numerical computations account for the detailed droplet shape, the repulsion force can be computed in a simplified manner via representing the droplets by point dipoles. At this point we restrict ourselves to the fundamental assumptions and the final outcomes of this approach, and refer to the Supplementary Methods for a rigorous derivation and discussion. The electrostatic problem can be recast into an idealized problem of two interacting bodies that are fused mirror-reflected spherical caps arranged in a parallel-plate capacitor with a plate separation of twice the value of the original design. Ignoring any higher-order multipole moments, the repulsion force can be approximated by half the force between two point dipoles induced at the center of the mirror-fused spherical caps

$$F_r = \frac{3}{8} \frac{p_1 p_2}{\pi \varepsilon_0} \frac{1}{d^4} \; , \qquad (3)$$

with the dipole moments[53]

$$p_1 = 4\pi \varepsilon_0 \Theta V_1 E_0 \left( \frac{1 - \Theta V_2/d^3}{1 - \Theta^2 V_1 V_2/d^6} \right) \; ,$$
$$p_2 = 4\pi \varepsilon_0 \Theta V_2 E_0 \left( \frac{1 - \Theta V_1/d^3}{1 - \Theta^2 V_1 V_2/d^6} \right) \; . \qquad (4)$$

Here, $E_0$ denotes the magnitude of the external electric field given by $-\Delta\phi/h$ (see Fig. 3a). The parameter $\Theta$ is a function of the apparent contact angle only and reads[54]

$$\Theta = \frac{24 \sin^3 \tilde{\theta}}{\pi \left( 2 - 3\cos\tilde{\theta} + \cos^3\tilde{\theta} \right)} \int_0^\infty \tau^2 \left( \frac{\tanh \pi\tau}{\tanh\left(\pi - \tilde{\theta}\right)\tau} - 1 \right) d\tau. \qquad (5)$$

The improper integral in the relation above can be evaluated numerically with little effort.

In the general case, i.e. without relying on the point-dipole model, the following scaling law can be derived for the repulsion force:

$$F_r = \text{fn}\left( \tilde{\theta}, \frac{V_1}{d^3}, \frac{V_2}{d^3} \right) \frac{\varepsilon_0 V_1 V_2 E_0^2}{d^4} \; , \qquad (6)$$

where in general the function $\text{fn}(\tilde{\theta}, V_1/d^3, V_2/d^3)$ is not known a priori and needs to be measured or computed. This scaling can be derived using a simple dimensional analysis. The governing parameters for the repulsion force are the contact angle $\tilde{\theta}$, the droplet volumes $V_1$ and $V_2$, the distance $d$, the permittivity of air $\varepsilon_0$ and the applied electric field strength $E_0$. Based on that, the following dimensionless groups can be identified

$$\Pi_1 = \tilde{\theta}, \quad \Pi_2 = \frac{V_1}{d^3}, \quad \Pi_3 = \frac{V_2}{d^3}, \quad \Pi_4 = \frac{F_r d^4}{\varepsilon_0 V_1 V_2 E_0^2} \; . \qquad (7)$$

Using the Buckingham $\Pi$ theorem, the repulsion force can be written in the form of Eq. (6).

**Magnitude of the repulsion force**. In Fig. 4a, the magnitude of the repulsion force $F_r$ is shown as a function of the distance between the two droplets $d$ for different parameter combinations $V_1$, $V_2$, and $\Delta\phi$. Experimental data are compared with the results of numerical simulations. As expected, the repulsion force for

zero applied electric field vanishes within the error bounds. Obviously, the experimental and numerical values agree very well. We emphasize that no fitting parameters are involved in the numerical computations. The figure clearly shows that the force decays rapidly with increasing $d$ as suggested by the scaling law Eq. (6).

As indicated by Fig. 4b, the interacting dipole model is suitable to predict the repulsion force with reasonable accuracy. For decreasing $d$ the contributions of the neglected higher-order multiple moments become more pronounced, leading to increasing deviations between the semi-analytical model and the numerical one.

From the scaling law Eq. (6) we conclude that as long as the droplets are far apart from each other, i.e. $d^3 \gg V_1, V_2$, the force should scale roughly as $V_1 V_2 E_0^2$ if $d$ is fixed. Obviously, our measured and computed data matches this scaling law quite well, as can be seen from Fig. 4c, even if the condition of large distance between the droplets is not strictly fulfilled.

Even though the droplets in the experiments described above have similar size, this should by no means imply that this a prerequisite for the principle to work. As shown in Supplementary Movie 3, repulsion can also be observed for droplets with different volumes. It immediately follows from Eq. (6) that for a fixed distance $d$ the repulsion force rapidly decreases when the droplets become smaller. On the other hand, when the droplet volume is reduced, smaller values of $d$ become relevant. Keeping in mind that the repulsion force rapidly increases when $d$ decreases, it depends on the scaling of the coalescence-promoting forces with $d$ and the droplet volume how efficient coalescence can be suppressed for smaller droplets.

**Complex multi-droplet dynamics**. As mentioned above, preventing droplets from coalescing might be crucial to ensure the functionality of certain applications and devices involving multiple droplets. In order to demonstrate that the mechanism proposed in this work is suitable to efficiently perform this task, we study the impact of the electric field on the dynamics of multiple interacting droplets. For this purpose droplets with a volume of 20 μL are manually dispensed on the LIS with zero inclination angle. Initially the droplets are distributed rather randomly and are in static equilibrium. Subsequently, the droplets are set in motion by a local air flow parallel to the plates. The air flow accelerates the droplets to a maximum speed of around 25 mm s$^{-1}$. These experiments are performed for zero as well as non-zero electric field. In the latter case the applied potential is 5.5 kV (see Supplementary Movies 4 and 5). Figure 5 shows image sequences extracted from the captured videos. These image sequences clearly reveal that coalescence can be suppressed by application of an electric field. When the electric field is switched off, rapid coalescence is observed, presumably promoted by the short-range capillary attraction between the droplets[40–42]. With electric field, however, not a single coalescence event is observed even for time intervals much larger than that covered by the image sequence of Fig. 5. With further increase of the air flow, coalescence events occurred even with the external electric field switched on. This indicates that the limits of the repulsion mechanism are reached. i.e. the attractive capillary force between droplets overcomes the repulsive electrostatic force. Presumably, on surfaces other than LIS where the short-range capillary attraction is absent, the limits the repulsion mechanism will be observed for more vigorous droplet collisions.

**Sampling from droplet arrays**. When a liquid surface is exposed to a sufficiently strong electric field, it assumes a cone-like shape from which a jet emerges[55]. A special case of this phenomenon is a sessile drop in a surface-normal homogeneous electric field.

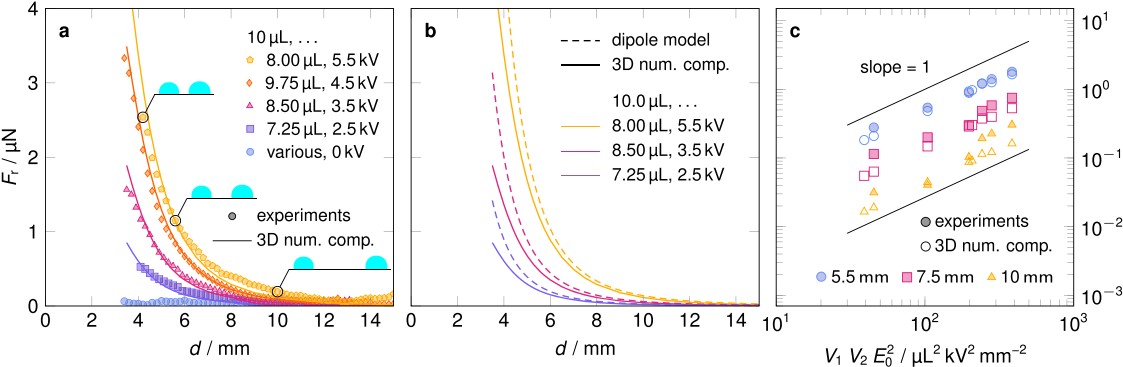

**Fig. 4 Repulsion force $F_r$ obtained from experiments, three-dimensional numerical computations and the semi-analytical model of interacting dipoles.** In all diagrams $F_r$ is plotted along the ordinate. **a** Repulsion force $F_r$ as a function of distance between the droplets. The filled symbols represent the averaged experimental data for different applied voltages $\Delta\phi$ and droplet volumes $V_1$ and $V_2$, respectively. Lines of the same color correspond to the numerical computations sharing the same parameters. At least six single experiments were averaged to obtain the data points. The insets show schematics of the droplet configuration for $d$ equal to 4.2, 5.6, and 10 mm, respectively. **b** The same as in **a**, but comparing the data of the numerical computations (solid lines) and the interacting dipole model (dashed lines). The colors indicate different parameter combinations. Using Mathematica[66] we found the integral in Eq. (5) to be 0.244 for a contact angle of 106°, hence $\Theta$ equals 0.589 for the present droplets. **c** Repulsion force $F_r$ as function of the product $V_1 V_2 E_0^2$ for three different values of $d$. The filled and empty symbols represent data from measurements and numerical computations, respectively. For fixed $d$ each data point corresponds to a different combination of the parameters $V_1$, $V_2$, and $\Delta\phi$. The solid straight lines with unity slope indicate the idealized linear scaling $F_r \sim V_1 V_2 E_0^2$. The maximum standard error of the mean force is less than 0.1 μN for all experimental data points. As a result, the corresponding error bars would be mostly smaller than the data symbols, which is why they are omitted.

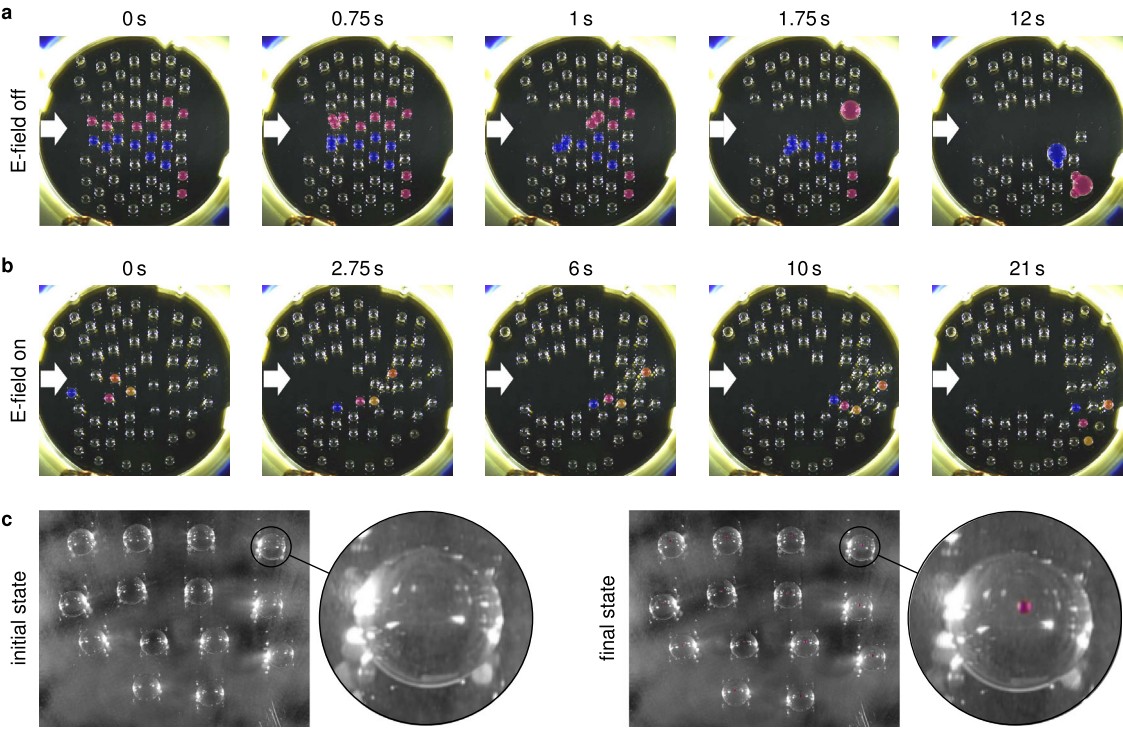

**Fig. 5 Multi-droplet dynamics: Coalescence suppression and sample withdrawal.** Multi-droplet dynamics in the parallel-plate capacitor induced by lateral air flow for **a** zero electric field and **b** an applied voltage of 5.5 kV with an air gap width of 3.5 mm. Colors were assigned to specific droplets by image post-processing. Droplets of the same color coalesce during the time interval covered by the image sequence. In part **b**, the coloring of droplets serves to track selected droplets. The white arrows indicate the direction of the lateral air flow and the position at which the air flow is applied. The time stamps of the frames indicate the time relative to the first frame of each sequence. **c** Parallel sampling from an array of droplets. The image on the left (right) side shows the array of 30 μL droplets before (after) the electric field strength was increased above the critical value. When the images were recorded, no field was applied. Small pendant droplets at the upper glass surface are visible in magenta. For better visibility the images were converted to grayscale format except for the small droplets. The magnified views show one specific droplets.

This situation was studied based on numerical simulations and experiments[56], and the critical field strength above which the liquid surface becomes unstable was determined. In the experiments reported above, the electric field strength was subcritical, such that coalescence is avoided via repulsive electrostatic interactions, but no instabilities are observed. The purpose of this subsection is to discuss the phenomena occurring at supercritical values of the field strength.

Droplets with a volume of 30 μL were dispensed on the LIS with zero inclination angle. Starting with a subcritical external electric field strength $E_0$ of 0.57 kV mm$^{-1}$, the electrostatic Maxwell stresses, balanced by gravity and surface tension, deform the initially oblate sessile droplets to a stationary prolate shape. If the electric field strength is increased to a supercritical value ($E_0 = 0.86$ kV mm$^{-1}$), this balance becomes unstable and Taylor cones emerge at the north poles of the droplets. The supercritical field strength was applied for about 6 s. Tiny secondary droplets detach from the cusps of the Taylor cones and are accelerated towards the upper electrode. To visualize these droplets, the surface of the upper fused-silica wafer facing the LIS was powdered with rhodamine B prior to the experiments. Rhodamine B changes its color from dark green to magenta if dissolved in water. Through the color change it becomes possible to visualize those regions in contact with liquid. The results of such an experiment are shown in Fig. 5c. After exceeding the critical field strength, magenta-colored droplets appeared at the upper surface. Based on the contact angle of water on a powdered fused-silica surface and the diameter of the magenta-colored regions, the volume of such a droplet is estimated as 16 nl. Supplementary Movie 6 shows a time sequence of the corresponding sample transfer in true color.

The underlying mechanism of the sampling mode described above that leads to the detachment of minute secondary droplets is a type of electrohydrodynamic spraying termed *micro dripping mode*[57]. This mode is characterized by the emission of single droplets from a liquid apex, in contrast to the decays of a liquid jet. Figure 5c indicates that the sample transfer occurs in a very similar manner for each of the parent droplets. Therefore, we have shown that minute samples can be withdrawn from an array of sessile droplets in a parallel manner, without interaction effects between neighboring droplets compromising the transfer process.

By reducing the distance between the two surfaces from 7 to 4 mm, fixing the electric field strength to roughly $E_0 = 1.5$ kV mm$^{-1}$ and applying 15 μL droplets on the LIS, a second sampling mode could be identified. In this mode, almost entire droplets (with the exception of very small residues) could be transferred from the LIS to the upper surface. The underlying mechanism is based on the deformation of the initially oblate droplets by the electrostatic stress into a prolate shape. The Maxwell stress stretches the droplets until they finally touch the upper surface. Since the adhesion between the rhodamine B powdered glass surface and a droplet (equilibrium contact angle $(67 \pm 3)°$) is substantially larger than that between the LIS and a droplet (equilibrium contact angle $(103 \pm 1)°$), the droplet is entirely transferred to the upper surface.

## Discussion

In summary, we studied the behavior of droplets on LISs under homogeneous electric fields. The electric field serves to suppress droplet coalescence, i.e. approaching droplets are repelled from each other. The repulsion force acting on the droplets can be clearly attributed to interactions of induced electric dipoles rather than native charges of the droplets. The experimentally and numerically determined repulsion forces are in very good agreement. Moreover, a semi-analytical interacting dipole model provides easy access to the repulsion force with decent accuracy. Different from standard schemes for suppressing coalescence based on surfactants, our approach offers real-time coalescence control by switching the electric field on an off. When the electric field strength exceeds a critical value, Taylor cones develop at the north poles of the droplets from which minute daughter droplets pinch off. This phenomenon can be utilized for withdrawing samples from an array of droplets in a parallel manner.

We believe that because of the real-time control of coalescence in complex droplet flows together with the option of parallel sampling from droplet arrays, this type of droplet microfluidics offers opportunities going beyond existing implementations. Combining these functionalities with emerging approaches such as propelling droplets on LIS using surface acoustic waves[44] or droplet transport on two-phase LIS[45] could enable a wide range of different applications.

## Methods

In the following, several aspects related to the experiments as well as details of the numerical model are presented. Further information can be found in the Supplementary Information.

**Fabrication of the LIS**. Following the method proposed in ref. [50] and utilized in ref. [43], the LIS was fabricated starting with a single-side polished silicon wafer (100 mm diameter, 525 μm thickness), sputter-coated with a film of aluminum about 100 nm thin. While submerged in low viscosity trimethyl-terminated poly-dimethylsiloxane (5 cSt, Sigma Aldrich), commonly known as silicone oil, the wafer was heated to 300 °C for 60 min. The silicone oil reacts with the hydroxy groups of the substrate[58] and forms a covalently bound thin film[59–62]. Subsequently, the wafer was cleaned by ultrasonication in isopropanol for 10 min, rinsed with DI water and dried with nitrogen. After that, the wafer was spin coated for 60 s at 1000 rpm with silicone oil of the same type resulting in a homogeneous layer less than 3 μm thick. This cleaning and impregnating procedure could be repeated over 60 times without changing the apparent dynamic and equilibrium contact angles of a DI water droplet on the LIS within uncertainty limits (see Supplementary Methods). For each measurement a freshly cleaned and impregnated wafer was used.

**Applying the electric field**. The aluminum coating of the silicon wafer was connected to ground of the high-voltage source (Heinzinger PNC 6000–100) via the clamping ring, while the ITO coating on the transparent fused-silica wafer (100 mm diameter, 0.5 mm thickness, Siegert Wafer) was connected to the high-voltage output with copper tape. To avoid destruction by accidentally triggered electric arcs, the sensitive ITO coating faced towards the camera such that there was an insulating glass layer between the electrodes. Using a 3D printed polyamide circular spacer the air-filled gap between the opposing wafers was fixed to 10 mm for the repulsion force measurements.

After charging the parallel-plate capacitor no current could be measured. Numerical computations indicate that the maximum local strength of the electric field around the droplets is less than half of the dielectric strength of air (3 kV mm$^{-1}$), based on the parameters prevailing in the repulsion force measurements. This supports the assumption that any charge transport due to ionization of air can be neglected. Furthermore, electrohydrodynamic phenomena like Taylor cones, tip streaming or liquid bridges did not occur for the subcritical electric field strengths applied in the repulsion force measurements. Supercritical external electric field strengths were applied intentionally for the sampling from droplet arrays. Here, the air gap width was reduced to 7 mm by using a thinner spacer such that the maximum voltage of 6 kV provided by the voltage source was sufficient to trigger tip streaming.

In all experiments we ensured that the droplets were always sufficiently far away from the edge of the parallel-plate capacitor such that potential edge effects can be neglected in our considerations.

**Camera adjustment and calibration**. Both the capacitor and the camera were mounted on tilting stages that allow for precise adjustments. The optical axis was aligned orthogonally to the capacitor plates utilizing a digital level, a laser mirror system and a 90° calibration angle (DIN875/2). A calibration microscopy slide (Bresser) placed onto the aluminum coating of the silicon wafer and aligned with the camera sensor was used for geometric camera calibration. The resulting calibration constant that relates the pixel to real world coordinates is 40.8 μm px$^{-1}$ for the experiments shown in Fig. 4.

**Volume of the mobile droplets**. The volume of the immobilized droplet could be regarded as given because it was applied with an Eppendorf pipette (statistical error: 1.45 %) prior to the measurements. On the other hand, the volume of the mobile droplets depended heavily on the pumping pressure, the inclination, the applied voltage, and the nozzle diameter, and could not be adjusted precisely with our setup. It rather fluctuated stochastically with large variance. Thus, the volume had to be deduced from the diameter of the droplet's visible projection during post-processing of the recorded images. This requires information about how the projection's diameter, the droplet volume as well as the applied voltage relate to each other. Comparison between sessile and sliding droplets of the same volume revealed that their projected diameters differ by less than one pixel. Hence, it is justified to ignore any deformation of the droplets due to motion and to perform the corresponding calibration measurements on LIS with zero inclination. Further details and results are given in the Supplementary Methods.

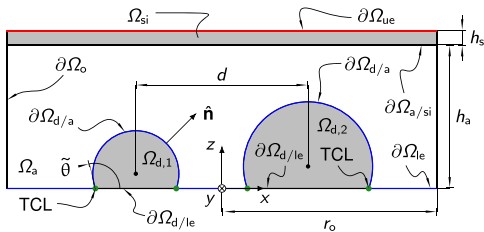

**Fig. 6 Schematic cross-sectional view of the three-dimensional computational domain with all relevant dimensions.** The domain is cut along the symmetry $xz$-plane. Domains and boundaries are denoted as $\Omega$ and $\partial\Omega$, respectively. The idealized three-phase contact lines (TCL) of both droplets are highlighted by green dots. Between the boundaries $\partial\Omega_{\text{ue}}$ (red) and $\partial\Omega_{\text{le}} \cup \partial\Omega_{\text{d/a}}$ (blue) a potential difference $\Delta\phi$ is applied. The sketch is not to scale, in particular the outer boundaries $\partial\Omega_{\text{o}}$ are actually placed far away from the droplets (see also Fig. S7).

**Validation of the friction law**. To validate the friction law stated in Eq. (1), we recorded the quasi-steady sliding velocity of DI water droplets pipetted manually on fresh LIS. Here, the droplet volume and inclination of the LIS were varied in the ranges 6–30 μL and 6.5–15°, respectively. Resulting plots showing the drag force as function of the velocity can be found in the Supplementary Methods. In summary, for DI water droplets slower than 20 mm s$^{-1}$ the friction law is valid. To account for that we fixed the inclination $\alpha$ to 7.5° and restrict our analysis to mobile droplets smaller than 11 μL. Droplets smaller than 5 μL could not reliably overcome pinning forces and were excluded from the analysis as well.

**Electrostatics**. In Fig. 6 schematic of the computational domain is provided for purpose of illustration. All involved dielectric media are assumed to be linear, isotropic and homogeneous. Since the relative permittivity of water is about 80 times higher than that of air, the major voltage drop is expected to occur across the air gap, while the electric field within the droplets is negligible. While this already applies to water droplets of zero conductivity, when ions are dissolved in the droplets they become conductive and shield their interior from electric fields. Thus in the framework of electrostatics it is justified to assume perfectly conducting droplets. As stated above, the thin lubricant film cloaking the droplet is neglected in the numerical computations. From these two assumptions, it immediately follows that the droplet/air interface $\partial\Omega_{\text{d/a}}$ is at the same potential as the lower electrode. Consequently, we impose a potential difference of $\Delta\phi$ between the upper electrode $\partial\Omega_{\text{ue}}$ and the effective lower electrode $\partial\Omega_{\text{le}} \cup \partial\Omega_{\text{d/a}}$. Since the induced surface charges on either side of the idealized droplet footprint compensate each other (see Fig. 3a) this surface appears electrically neutral from a macroscopic point of view, and the droplets $\Omega_{\text{d,1}}$ and $\Omega_{\text{d,1}}$ can be subtracted from the computational domain. As a result, integrating the surface charge over the droplet/air interface $\partial\Omega_{\text{d/a}}$ yields a non-zero apparent net charge. The electric field is expected to converge asymptotically to $\mathbf{E} = E_0\hat{\mathbf{z}}$ with increasing distance from the droplets. This translates into homogeneous Neumann conditions $\nabla\phi \cdot \hat{\mathbf{n}} = 0$ at the vertical boundaries $\partial\Omega_{\text{o}}$ arranged far away from the droplets ($r_o \gg h_a$).

We further exploit the inherent reflection symmetry of the electrostatic potential around the interacting droplets with respect to the $xz$-plane and substitute one half of the three-dimensional domain by an appropriate symmetry condition, more specifically $\nabla\phi \cdot \hat{\mathbf{n}} = 0$. At the interface between air and the fused-silica surface $\partial\Omega_{\text{a/si}}$ the common conditions for interfaces with zero free surface charge apply: $[\![\mathbf{D}]\!] \cdot \hat{\mathbf{n}} = 0$ and $[\![\mathbf{E}]\!] \cdot \hat{\boldsymbol{\tau}}_i = 0$. Here, $\mathbf{D} = \varepsilon_0\varepsilon_r\mathbf{E}$ and $\hat{\boldsymbol{\tau}}_i$ with $i = 1, 2$ denote the displacement field and the tangential unit vectors, respectively. The relative permittivity of fused-silica is set to 3.8[63]. To a good approximation, the permittivity of air is equal to the vacuum permittivity $\varepsilon_0$, i.e. $\varepsilon_r = 1$. Because of the relatively small potential drop in the fused-silica wafer, we conclude that the magnitude $E_0$ of the applied electric field $\mathbf{E}_0 = E_0\hat{\mathbf{z}}$ can be estimated by $-\Delta\phi/h_a$. The height of the air gap $h_a$ and the thickness of the fused-silica wafer $h_{\text{si}}$ are the same as in the experiments (see above).

We note that in the limit $\varepsilon_d \to \infty$ the induced bound surface charges $\sigma_b = [\![\mathbf{P}]\!] \cdot \hat{\mathbf{n}}$ of a perfect dielectric exactly resemble the distribution of free surface charges $\sigma_f = [\![\mathbf{D}]\!] \cdot \hat{\mathbf{n}} = \mathbf{D} \cdot \hat{\mathbf{n}}$ of a perfectly conducting body. Thus, in these two cases the forces exerted on the body are identical.

Instead of integrating the electrostatic Maxwell stress according to Eq. (2) the net repulsion force can also be computed utilizing the virtual work principle. This alternative approach, which is presented in the Supplementary Methods in more detail, yields equivalent results.

The governing equations and corresponding boundary conditions were implemented and solved in the proprietary finite-element framework COMSOL Multiphysics®[64]. Second-order Lagrangian elements were used to spatially discretize the scalar field $\phi$. Mesh convergence was studied thoroughly to ensure that the presented solutions are mesh-independent with respect to appropriate tolerances (see the Supplementary Methods for further details). The system of

linear equations resulting from spatial discretization was solved iteratively using a multigrid preconditioned conjugate gradient method.

**Repulsion force between deformed droplets**. Evidently, Eq. (2) implies that the repulsion force resulting from the electrostatic field depends on the shape of the droplets. In general, the shape is governed by a dynamic balance of surface tension, electrostatic Maxwell stresses, hydrostatic pressure due to gravity as well as hydrodynamic stresses originating from flow within the different phases. Since the droplets are coupled by the electric field, a change of the interface shape of either droplet induces a change of the other droplet's interface and vice versa. However, from experimental observations and preliminary numerical computations we infer that in the present case the droplet shapes differ only little from axisymmetric static equilibrium configurations. Hence, it seems reasonable to ignore any contributions entering this rather complex balance that induce motion and prevent a static equilibrium from being established. By doing so we pretend that the droplets are at rest and that the electric field each droplet is exposed to is not perturbed by the other droplet. The surface tension, the hydrostatic pressure due to gravity, and the electrostatic Maxwell stress enter the modified Young–Laplace equation that determines the droplet shape:

$$\Delta p = \tilde{\gamma}\,\kappa + \Delta\rho\,g\,z - \frac{1}{2}\varepsilon_0(\mathbf{E}\cdot\mathbf{E}).\qquad(8)$$

Here, $\tilde{\gamma}$ and $\kappa = \nabla \cdot \hat{\mathbf{n}} = 1/R_1 + 1/R_2$ denote the effective surface tension and local curvature of the idealized droplet/air interface $\partial\Omega_{\text{d/a}}$, respectively. Evidently, in turn the local distribution of the electric field $\mathbf{E}$ at the droplet surface depends heavily on the shape of the latter. Determined by the constraint of constant volume, the pressure difference $\Delta p$ can be interpreted as a Lagrangian multiplier. Along the three-phase contact line the static contact angle $\tilde{\theta}$ is imposed to close the problem and ensure a unique solution.

Instead of solving the static Young–Laplace equation explicitly, we compute the coupled time evolution of the flow field inside the droplet and the electrostatic field around it simultaneously until a static equilibrium is reached. As initial condition perfectly spherical droplets are imposed. Electrostatics and fluid dynamics are two-way coupled by a dynamic stress balance at the deforming droplet/air interface which simplifies to the Young–Laplace equation in equilibrium. Further information on this implicit solution approach can be found in the Supplementary Methods.

The interfacial tensions of the lubricant/air $\gamma_{\text{la}}$ and lubricant/droplet $\gamma_{\text{ld}}$ interface are found to be 19.7[43] and 39.7 mN m$^{-1}$[65]. Thus the resulting effective surface tension $\tilde{\gamma} = \gamma_{\text{ld}} + \gamma_{\text{la}}$[37] is 59.4 mN m$^{-1}$. Using a Krüss Drop Shape Analyzer DSA100 the apparent contact angle was measured to be around 106° (see also the Supplementary Methods), which is consistent with the theoretical approximation $\cos\tilde{\theta} = (\gamma_{\text{la}} - \gamma_{\text{ld}})/\tilde{\gamma}$ proposed in [37]. In the Young–Laplace equation (8) we implicitly assumed the gravitational acceleration $\mathbf{g}$ to be purely normal to the electrodes, such that a static equilibrium can establish. Obviously and by contrast, in the experiments the tangential component of the gravitational acceleration is sufficient to induce motion. Nevertheless, the normal acceleration $\mathbf{g} \cdot \hat{\mathbf{z}} = g\cos\alpha$ can be approximated by $g$ since the inclination angle $\alpha$ is small. The density difference between the droplet and the ambient air $\Delta\rho$ is approximated by the density of water 997 kg m$^{-3}$.

In total, our computational procedure to determine the repulsion force is composed of two successive steps. First, we compute the static equilibrium shape for each droplet individually as described above. Since the stress distribution at the interface is axisymmetric with respect to the vertical axis, we can employ cylindrical coordinates. Second, we compute the three-dimensional electrostatic field around two rigid droplets of given volumes for different distances $d$. The shape of each droplet is extracted from the static equilibrium computed in the previous step. Eventually, the repulsion force is inferred from the electric field distribution according to Eq. (2). We note that for the parameter ranges covered in our experiments the electrostatic Maxwell stress partially compensates the hydrostatic pressure, such that the computed equilibrium shapes differ only little from ideal spherical caps (see Fig. S4) predicted by the Young–Laplace equation if only capillary forces are considered, i.e. $\Delta p = \tilde{\gamma}\,\kappa$. In general, this might not be the case which is why we propose this approach if the shape of the droplets is not known a priori. Inevitably, the repulsion force obtained by means of this purely static model is subject to certain errors, as the we ignore any stress contributions that result in non-axisymmetric droplet shapes. We refer to the Supplementary Methods for a more detailed discussion of the non-axisymmetric components of the electrostatic Maxwell stress. In fact the described model seems to represent a good compromise between computational efficiency and inclusion of details.

## Data availability

Source data of diagrams shown in Figs. 2a–c, 3b, 4a–c, Supplementary Figs. 2a–b, 3, 4, 5, and 7a–b are available at https://doi.org/10.48328/tudatalib-675. Any additional data generated and/or analyzed as part of the present work are available from the corresponding author on reasonable request.

## Code availability

Custom computer code used to analyze experiments is available from the corresponding author on reasonable request. The same applies to the numerical model implemented in COMSOL Multiphysics®.

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

## Acknowledgements
We kindly acknowledge the financial support of J.H. by the German Research Foundation (DFG) within the Collaborative Research Centre 1194 "Interaction between Transport and Wetting Processes".

## Author contributions
J.H., M.T.S. and S.H. designed research; J.H. performed experimental research; M.T.S. performed numerical computations; J.H. and M.T.S. analyzed data; J.H., M.T.S. and S.H. wrote the paper.

## Funding

## Competing interests
The authors declare no competing interests.
