## [Peer Review File · Nature Communications]

Manipulation and control of droplets on surfaces in a homogeneous electric fieldREVIEWER COMMENTS

Reviewer #1 (Remarks to the Author):

It is an in-depth manuscript demonstrating the droplet manners on a liquid-infused surface in a homogenous electric field and deeply elaborating the numerical computations during the electrically influenced common droplet motion process. First, authors did a clear work on describing the trajectories and figuring the friction law of droplet when pushed away from the immobilized droplet on a LIS in the homogeneous electric field. The calculating process was quite convincing. Second, the explanation about the repulsion force between droplets attributing to the Coulomb interaction of surface charges induced by the external electric field made sense, which focused on the asymmetric Maxwell stress distribution at different sides of a droplet. There are still questions not figured out in this manuscript. How to measure the net charges in single water droplet? When conducting multi-droplet dynamics, how to assure that the repulsion force is sufficient to avoid droplet coalescence due to the air flow? The authors tracked merging droplets through coloring, but how to accurately design which droplets to be manipulated by applied electric field beforehand? Small pendant droplets sampling from an array of droplets is an interesting work, could the content and volume of the isolated one be controlled by the electric field? If so, this could be a versatile method in medical extraction, especially applied in vaccines which benefit for this era of epidemic. All the implicit deductions and computations on mathematical and physical models were explained well in detail in this manuscript, however, similar experimental phenomenon about surface charge induced droplet manipulation on solid surface has been published by other groups, such as papers *Nat. Mater.* 18, 9336-941, 2019 and *Adv. Mater.* 31, 1905449, 2019. The relevant references should be cited in this manuscript. The experimental results in this manuscript are suitable for a comprehensive magazine as *Nat. Commun.*. If the authors could consider my opinions and advices, and made corresponding improvement on the draft, I would be willing to recommend publication of this manuscript.

Reviewer #2 (Remarks to the Author):

This is an excellent paper that is well written and of interest to anyone working in or near the fields of droplets on surfaces, LIS/SLIPS or Microfluidics. The conclusions are well supported by the data and the analysis is impeccable.

I would recommend publication in Nature communications.

Reviewer #3 (Remarks to the Author):

Hartmann et al. propose an electric field-induced strategy to manipulate droplets on liquid-infused surfaces. Electric dipoles are induced inside the droplets, which generate repulsive dipole force to suppresses the coalescence of droplets. The parameters affecting the repulsive force, including the droplet volumes, the droplets distance, and the electric field strength, are investigated using experimental, numerical computational, and semi-analytical methods. However, the novelty of this work is not clear, and the current manuscript cannot meet the high standard for Nature Communications.

General comments.

1. What is the main finding of this work? The droplet surface charging and ions redistribution in the electric field are common phenomena, such as *Advances in Colloid and Interface Science*, 2016, 236, 142-151; *Nature Communications*, 2013, 4, 2517. The suppression of droplet coalescence by droplet charging also has been reported *Scientific Reports*, 2013, 3, 2037.
2. In the title of this manuscript, what is the difference between the words "manipulation" and "control"?
3. The states that "the thin lubricant film that cloaks the droplet and forms a wetting ridge around the footprint is ignored in our electrostatic model since its contribution is marginal" should be discussed in detail.
4. For the computed Maxwell stress in Fig. 3b, it is suggested to provide a clear demonstration of the asymmetric distribution.

5. The "sampling from droplet arrays" section seems confusing in the text. What is the connection between this section and the formers? The droplet manipulation mechanism differs a lot. In addition, electrohydrodynamic jetting has been well studied and there is nothing new.
6. The existence of the oil layer wrapping the droplet is trouble for further manipulations. How to separate the droplet from the oil layer?
7. Will the oil permeate into and contaminate the droplet especially the droplet has complex ingredients?
8. It is suggested to add corresponding descriptions for the videos.
9. The manipulation of multi-droplets section is interesting, and it is appreciated to add more discussion and explorations.

Reviewer #4 (Remarks to the Author):

In this study, the authors investigate droplet (water) control and motion in a homogeneous electric field. The study was well-planned, and the manuscript is well-organized and written. The authors used a liquid-infused surface in an inclined setup, and they present that the repulsive dipole force suppresses the coalescence of droplets moving on a liquid-infused surface. The experimental setup is specific and very limited (inclined setup, special surface treatment, electric field, the droplet size is around a few millimeters). I do not see how this method could be used in a broad range of real applications. This investigation would draw attention to the research community working on self-propelled motion and droplet manipulation. Therefore, I recommend rejecting the paper and transferring it to a more specialized journal (Communications in Physics or Scientific Reports).

The authors might improve the quality of their presentation by addressing the following issues:

- (i) Please discuss the effect of the droplet size (e.g., interaction of two droplets of different sizes). What is the size limit of the droplet to experience this phenomenon?
- (ii) Did the authors try to use polar protic and aprotic solvents instead of using water? What would be the effect?
- (iii) How might the ionic strength affect the phenomenon observed?

In the following, the reviewers' comments are typeset in italics while our replies appear in regular font.

Reviewer #1 (Remarks to the Author):

It is an in-depth manuscript demonstrating the droplet manners on a liquid-infused surface in a homogenous electric field and deeply elaborating the numerical computations during the electrically influenced common droplet motion process. First, authors did a clear work on describing the trajectories and figuring the friction law of droplet when pushed away from the immobilized droplet on a LIS in the homogeneous electric field. The calculating process was quite convincing. Second, the explanation about the repulsion force between droplets attributing to the Coulomb interaction of surface charges induced by the external electric field made sense, which focused on the asymmetric Maxwell stress distribution at different sides of a droplet.

We thank the reviewer for her/his positive feedback regarding the profoundness of our work.

There are still questions not figured out in this manuscript. How to measure the net charges in single water droplet?

As mentioned in our manuscript (see section "Origin of the repulsion force") we cannot rule out that the droplets carry a net charge before being deposited on the liquid-infused surface. However, we also explain why the observed repulsion is independent of the existence or non-existence of these charges. In our opinion, there is no need to measure the net charges, since they do not contribute to the repulsion force. In a general context, one could measure them as suggested by Choi et al. (Scientific Reports 3, 2019, DOI: 10.1038/srep02037), for example.

When conducting multi-droplet dynamics, how to assure that the repulsion force is sufficient to avoid droplet coalescence due to the air flow?

As a matter of fact, there is a maximum air flow rate at which coalescence is still suppressed. Above that flow rate, coalescence sets in. This fact is briefly discussed in the section "Complex multi-droplet dynamics" of the revised version. However, we do not attribute too much significance to the exact conditions under which coalescence sets in, since we view the principle described in our paper as a general principle that can not only be employed in context with LIS. It is well conceivable to control the coalescence of droplets inside microchannels with asymmetric dielectric layout based on the same principle. We have chosen to work with droplets on LIS mainly for two reasons: 1) LIS provide well-defined conditions to study droplet trajectories; 2) Owing to the short-range attractive force between droplets (Jiang et al., PNAS 116, 2019, DOI: 10.1073/pnas.1817172116; Sun & Weisensee, Soft Matter 15,

2019, DOI: 10.1039/C9SM00493A; Kajiyama et al., *Soft Matter* 12, 2016, DOI: 10.1039/C6SM01883A; all cited in our manuscript), suppressing coalescence becomes especially challenging. For example, had we worked with droplets in a Hele-Shaw cells with oil as continuous phase, there would have been a short-range dissipative force counteracting coalescence (due to oil-film squeeze out), in contrast to our case where we have a short-range force promoting coalescence. Therefore, and because of the broad applicability of our principle of coalescence control, we do not attribute too much significance to the exact conditions under which the electrostatic repulsion becomes too weak to suppress coalescence.

The authors tracked merging droplets through coloring, but how to accurately design which droplets to be manipulated by applied electric field beforehand?

Our current setup does not allow to suppress coalescence of specific droplets while allowing it for others. Within the homogeneous external electric field between the two plate electrodes, all droplets are repelled from each other. However, one could easily think of defining regions with and without electrodes in a parallel-plates geometry in order to locally control coalescence. By the way, the coloring was added during the post-processing of images.

Small pendant droplets sampling from an array of droplets is an interesting work, could the content and volume of the isolated one be controlled by the electric field? If so, this could be a versatile method in medical extraction, especially applied in vaccines which benefit for this era of epidemic.

Controlling the volume of the samples is surely a topic of significant interest for practical applications. Coming up with a comprehensive answer, however, is very challenging since we have a vast number of parameters (distance between the plates, electric field strength, size of the primary droplet, thickness of the oil film, wettability of the upper surface etc.) that govern the sampling process. So far we have identified two different modes of sample transfer. The first mode is the one that was already described in the original version of the paper. In the second mode, entire droplets are transferred from the lower to the upper surface. This mode is now also described in the revised version of the paper, see the section "Sampling from droplet arrays".

*All the implicit deductions and computations on mathematical and physical models were explained well in detail in this manuscript, however, similar experimental phenomenon about surface charge induced droplet manipulation on solid surface has been published by other groups, such as papers *Nat. Mater.* 18, 9336-941, 2019 and *Adv. Mater.* 31, 1905449, 2019. The relevant references should be cited in this manuscript.*

As discussed in the section "Origin of the repulsion force", the mechanism proposed by us relies solely on the induction of electric dipoles by the external electric field. It

neither depends on the interaction of droplets with surfaces whose charge is controlled based on a specific chemistry (Sun et al., Nature Materials 9, 2019, DOI: 10.1038/s41563-019-0440-2) nor on net charges of the droplets (Dai et al., Advanced Materials 31, 2019, DOI: 10.1002/adma.201905449). By switching the external electric field on and off, droplet coalescence can either be suppressed or allowed, which represents a significant advantage over principles that rely on irreversible changes of the surface chemistry. Since in our case the repulsion is largely independent of the surface chemistry, our mechanism should also work on surfaces other than LIS, which is a major benefit. In the introductory section of the revised version, we decided to highlight this crucial difference between our work and others more prominently.

The experimental results in this manuscript are suitable for a comprehensive magazine as Nat. Commun.. If the authors could consider my opinions and advices, and made corresponding improvement on the draft, I would be willing to recommend publishment of this manuscript.

We hope that the answers and clarifications given above will convince the reviewer to recommend publication in Nature Communications.

Reviewer #2 (Remarks to the Author):

This is an excellent paper that is well written and of interest to anyone working in or near the fields of droplets on surfaces, LIS/SLIPS or Microfluidics. The conclusions are well supported by the data and the analysis is impeccable.

I would recommend publication in Nature communications.

We very much appreciate the reviewer's assessment concerning the quality of our work and her/his recommendation to publish it in Nature Communications.

Reviewer #3 (Remarks to the Author):

Hartmann et al. propose an electric field-induced strategy to manipulate droplets on liquid-infused surfaces. Electric dipoles are induced inside the droplets, which generate repulsive dipole force to suppresses the coalescence of droplets. The parameters affecting the repulsive force, including the droplet volumes, the droplets distance, and the electric field strength, are investigated using experimental, numerical computational, and semi-analytical methods. However, the novelty of this work is not clear, and the current manuscript cannot meet the high standard for Nature Communications.

Concerning the reviewer's claim of lacking novelty we would like to remark that from a fundamental point of view, inducing electric dipoles in droplets that cause a repulsion force is *per se* nothing new. Our claim of novelty, however, is that based on this principle, the coalescence between droplets can be controlled. In this context, we see our experiments performed on LIS only as a proof-of-principle for a scheme that could be applied much more widely. We hope that the following replies will convince the reviewer to reconsider her/his recommendation.

General comments.

1. *What is the main finding of this work? The droplet surface charging and ions redistribution in the electric field are common phenomena, such as Advances in Colloid and Interface Science, 2016, 236, 142-151; Nature Communications, 2013, 4, 2517. The suppression of droplet coalescence by droplet charging also has been reported Scientific Reports, 2013, 3, 2037.*

The main finding of our work is that homogeneous electric fields applied to sessile drops can be utilized to control their coalescence. Taking the applied voltage as a control parameter, we can induce different scenarios, from non-coalescence to coalescence.

The work by Li & Li (Advances in Colloid and Interface Science 236, 2016, DOI: 10.1016/j.cis.2016.08.006) is a purely theoretical paper that considers droplets of low permittivity in a high-permittivity medium, the opposite of our case. We do not see many similarities to our work.

Miljkovic et al. (Nature Communications 4, 2013, DOI: 10.1038/ncomms3517) report that tiny droplets jumping of a superhydrophobic surface repel each other due to net charges gained by charge separation at the substrate. This work relies on a specific surface chemistry and lacks the controllability via an external parameter (voltage) we are reporting.

Similar arguments apply with respect to the paper by Choi et al. (Scientific Reports 3, 2013, DOI: 10.1038/srep02037; cited in our manuscript), where it is shown that water droplets dispensed in oil repel each other. Also here, the control parameter that governs droplet coalescence is missing.

In order to avoid further misunderstandings, we decided to point out this fundamental difference in the introduction as well as the section “Origin of the repulsion force”.

2. *In the title of this manuscript, what is the difference between the words “manipulation” and “control”?*

We view this as follows: “Manipulation” refers to inducing a behavior that would not be observed in the absence of an electric field, such as the sampling from droplet arrays we are reporting. “Control” refers to influencing a behavior (via an external parameter) that is characteristic for droplets even without electric field, such as coalescence or non-coalescence.

3. *The states that “the thin lubricant film that cloaks the droplet and forms a wetting ridge around the footprint is ignored in our electrostatic model since its contribution is marginal” should be discussed in detail.*

A more detailed explanation and justification of this simplification is given in our previous work (Sinn et al., Applied Physics Letters 114, 2019, DOI: 10.1063/1.5091836), which we have cited in the original version of our manuscript. Due to limited space we refrain from recapitulating these arguments.

4. *For the computed Maxwell stress in Fig. 3b, it is suggested to provide a clear demonstration of the asymmetric distribution.*

In Fig. 5 in the SI of the original manuscript, the Maxwell stress distribution is shown exemplarily for different droplet separation distances. Especially for small distances the asymmetry becomes apparent.

5. *The “sampling from droplet arrays” section seems confusing in the text. What is the connection between this section and the formers? The droplet manipulation mechanism differs a lot. In addition, electrohydrodynamic jetting has been well studied and there is nothing new.*

We acknowledge that in terms of the physics, the sampling operation is only weakly connected with the rest of the paper. However, our intention was to demonstrate that the setup we are using (droplets on a LIS in a homogeneous electric field) enables a number of different useful operations, in the spirit of a proof-of-concept. Demonstrating the spectrum of operations enabled by a specific principle/device/structure is not uncommon in manuscripts published in the family of Nature journals, so we decided to include the sampling from droplet arrays as well.

We do not claim that we studied electrohydrodynamic jetting for the first time. Rather than that, the novelty of our work lies in the parallel sampling from a multitude of droplets, a process that has not been reported in the literature, to the

best of our knowledge. In our opinion, being able to draw samples in a parallel process from a droplet array is a major step forward from the viewpoint of applications, since in an application context (e.g. in pharmaceutical research) processes involving only a single droplet are hardly of any relevance.

6. The existence of the oil layer wrapping the droplet is trouble for further manipulations. How to separate the droplet from the oil layer?

We are not aware that this problem has been addressed in the literature. However, at this point we would like to mention that there are versions of LIS in which the lubricant does not cloak aqueous droplets. An overview of different lubricants and their cloaking behavior is given in (Sett et al., ACS Applied Materials and Interfaces 9, 2017, DOI: 10.1021/acsami.7b10756). Therefore, in practical applications where the oil layer may cause difficulties in the further manipulation of droplets, one can resort to non-cloaking lubricants.

7. Will the oil permeate into and contaminate the droplet especially the droplet has complex ingredients?

Fortunately, silicone oil is virtually insoluble in water. Furthermore, it is quite inert, non-toxic and highly biocompatible. For these reasons silicone oil is routinely used as the continuous phase in droplet microfluidics inside microchannels, and a large number of protocols based on droplets with complex ingredients have been demonstrated. When a droplet on a LIS is cloaked by a silicone oil film, its fundamental chemical environment is the same as for microchannel-based microfluidics, and we have no reason to believe that in such a case contamination will occur. As a matter of fact, in (Woo & Butt, Angewandte Chemie International Edition 56, 2017, DOI: 10.1002/anie.201611277) and (Agrawal et al., ACS Applied Materials and Interfaces 11, 2019, DOI: 10.1021/acsami.9b08849) it was demonstrated that in droplets on LIS, chemical reactions involving organic compounds can be conducted.

8. It is suggested to add corresponding descriptions for the videos.

A description of the videos had already been included in the SI of the original version of our paper.

9. The manipulation of multi-droplets section is interesting, and it is appreciated to add more discussion and explorations.

In the revised version of the manuscript, we added a brief discussion of the limits of coalescence suppression in multi-droplet transport.

Reviewer #4 (Remarks to the Author):

In this study, the authors investigate droplet (water) control and motion in a homogeneous electric field. The study was well-planned, and the manuscript is well-organized and written. The authors used a liquid-infused surface in an inclined setup, and they present that the repulsive dipole force suppresses the coalescence of droplets moving on a liquid-infused surface. The experimental setup is specific and very limited (inclined setup, special surface treatment, electric field, the droplet size is around a few millimeters).

We believe that the statement concerning the inclination is based on a misunderstanding. Here, we want to stress that the inclination of the capacitor is solely for reasons of reliable and repeatable measurements of the repulsion force, without the need to open the device after each single experiment and dispensing new droplets on the liquid-infused surface. Using an inclined capacitor is the simplest approach we could imagine to apply a well-defined driving force (gravitational force) to the droplets. We did not intend to suggest this as a specific propulsion mechanism in an application context. With respect to standard mechanisms of droplet transport, we refer to the papers cited in the first paragraph of the introductory chapter of our manuscript.

The reviewer further alludes to the “special surface treatment”. Also here, our choice (LIS) was mainly driven by an effort to provide experimental conditions as well-defined as possible. LIS are very homogeneous, have a very small contact-angle hysteresis and are therefore well suited to quantitatively study drop motion along surfaces. Furthermore, since the focus of our work is the control of drop coalescence, LIS are a very appropriate choice as well. Owing to the short-range attractive force between droplets on LIS, suppressing coalescence becomes especially challenging. For example, had we worked with droplets in a Hele-Shaw cells with oil as continuous phase, there would have been a short-range dissipative force counteracting coalescence (due to oil-film squeeze out), in contrast to our case where we have a short-range force promoting coalescence.

In summary, we regard our work as a proof-of-principle of a very general concept of coalescence control. The specific choices (inclined setup, LIS) were made because of a number of requirements in terms of conducting convincing and quantitative experiments, but should by no means suggest that our general principle of coalescence control only works in such a context.

I do not see how this method could be used in a broad range of real applications. This investigation would draw attention to the research community working on self-propelled motion and droplet manipulation. Therefore, I recommend rejecting the paper and transferring it to a more specialized journal (Communications in Physics or Scientific Reports).

We cannot follow the reviewer's assessment concerning the thematic scope of Scientific Reports. We cite from the web page of the journal: "*Scientific Reports* is an open access journal publishing original research from across all areas of the natural sciences, psychology, medicine and engineering."

Concerning the broad interest of our work, we disagree with the reviewer. Here we also refer to our reply above. As a matter of fact, we have demonstrated a rather generic principle for coalescence control of droplets. In contrast to related work in which the manipulation of droplets relies on a very special surface chemistry (Sun et al., *Nature Materials* 9, 2019, DOI: 10.1038/s41563-019-0440-2), our principle does not require special surfaces. Even when resorting to LIS it should be taken into account that such substrates can be fabricated in many different ways (Chen et al., *Materials Horizons*, 2020, DOI: 10.1039/d0mh00088d) using a large number of different lubricants (Sett et al., *ACS Applied Materials and Interfaces* 9, 2017, DOI: 10.1021/acsami.7b10756). It is even conceivable that our principle of droplet manipulation can be applied to droplets inside microchannels with asymmetric dielectric layout. Therefore, we believe that our work will be of interest to a broad scientific community.

The authors might improve the quality of their presentation by addressing the following issues:
(i) *Please discuss the effect of the droplet size (e.g., interaction of two droplets of different sizes). What is the size limit of the droplet to experience this phenomenon?*

We have added information on the interaction of droplets of different sizes to the revised version of the manuscript, see the section "Magnitude of the repulsion force". We have also decided to include an additional video in the SI, showing the interaction of two droplets with different volumes. Concerning the general dependence of the repulsion force on the droplet size, we refer to Eq. 6. As indicated above, we view our principle of coalescence control as a rather general one that could be applied in a number of different contexts. Droplets in microchannels, for example, are usually smaller than those considered by us. Eq. 6 suggests that the repulsion force scales with the droplet volume, i.e. it rapidly decreases when the droplets become smaller. On the other hand, the force rapidly increases when the distance between two droplets decreases. Viewing this information in context with the non-electric forces between two droplets (referred to in our reply to the first question) that can either promote or counteract coalescence, it becomes very difficult to make general statements about the droplet size dependence of our principle of coalescence control, that is, the size limit below which the principle no longer works. We regard our results as a proof-of-principle of a concept that is –by construction– applicable in a much more general context that is described in our paper.

(ii) *Did the authors try to use polar protic and aprotic solvents instead of using water? What would be the effect?*

The lower the polarizability, which depends on the relative permittivity and the ionic strength, the higher the required electric field. Because of this, aqueous droplets are better suited for our principle than droplets from organic liquids. In addition, in the latter case it is more challenging to ensure the immiscibility of the droplet with the lubricant. For these reasons we have focused on aqueous droplets.

(iii) How might the ionic strength affect the phenomenon observed?

Here we refer to information already given in the original version of our manuscript. Both limiting cases of perfectly dielectric (zero ionic strength) and perfectly conducting (infinite ionic strength) aqueous droplets were studied in the numerical computations (see section "Perfect dielectric vs. perfectly conducting droplets" in the SI of the original manuscript). In summary, the difference in the force is less than 10% for the smallest distances between the droplets. Note that a real droplet will always range between these ideal limits. However, because of the autoionization of water and since we did not take any special measures to avoid absorption of CO₂, the droplets in our experiments could be rather close to the limit of perfect conductors. This is supported by the good agreement between the experimentally determined and numerically computed repulsion force (Fig. 4). We decided to shift the corresponding paragraph where we discuss this issue from the subsection "Electrostatics" in the "Methods" section to the "Origin of repulsion force" section.

REVIEWERS' COMMENTS

Reviewer #1 (Remarks to the Author):

Review of the manuscript "Manipulation and Control of Droplets on Surfaces in a Homogeneous Electric Field"

Thanks for your convincing answer to my concerns. I recommended the acceptance since the question are resolved.

Thanks for clarifying that the electrostatic repulsion force is sufficient to suppress coalescence even under a maximum air flow rate in the new version.

If we can predict the trajectory of specific single droplet under electric field in advance, we can manipulate the droplet toward particular direction. Under the circumstances, the electric field not only provides the electrostatic repulsion force but also set a definite direction of movement for specific single droplet.

Since the volume of samples is indeed one of contributing factors, the two different modes of sample transfer is essential in your revised manuscript.

Reviewer #3 (Remarks to the Author):

In the revised manuscript and the response letter, the authors claim that the main finding is the control of droplet behaviors using a well-known principle. On this basis, although the droplet dynamics (coalescence/non-coalescence) is interesting, without a clear demonstration of the potential applications, the manuscript is recommended to be published in a more specialized journal.

General comments:

1. More explorations of multi-droplets manipulation are suggested to be enhanced.
2. In Fig. 3b, it's hard to tell the asymmetric distribution of the Maxwell stress, which is important to explain the repulsive forces between the adjacent droplet.

Reviewer #4 (Remarks to the Author):

The authors revised the manuscript and improved the quality of their manuscript. However, the revised manuscript cannot meet the high standard of Nature Communications. Therefore, I do not support publication of this manuscript.

Replies to the comments by reviewer #3

Comment

More explorations of multi-droplets manipulation are suggested to be enhanced.

Reply

We agree that this is a very interesting topic. In order to significantly extend the discussion on multi-droplet dynamics, we would need to perform and analyze further experiments. We feel that this would run contrary to the timely publication of our manuscript. In addition, we were given a two-weeks deadline for resubmission. For these reasons, we refrained from conducting additional studies of the multi-droplet dynamics.

Comment

In Fig. 3b, it's hard to tell the asymmetric distribution of the Maxwell stress, which is important to explain the repulsive forces between the adjacent droplet.

Reply

We revised Fig. 3b. The revised figure shows the situation of two droplets with a smaller separation distance, where the asymmetric distribution of the Maxwell stress is more prominent than in the earlier version.